# HPHT-Treated Impact Diamonds from the Popigai Crater (Siberian Craton): XRD and Raman Spectroscopy Evidence

**Anatoly Chepurov [1], Sergey Goryainov [1], Sergey Gromilov [2], Egor Zhimulev [1], Valeriy Sonin [1], Aleksey Chepurov [1], Zakhar Karpovich [1], Valentin Afanasiev [1,\*] and Nikolay Pokhilenko [1]**

[1] V.S. Sobolev Institute of Geology and Mineralogy SB RAS, 3 Akademika Koptyuga, 630090 Novosibirsk, Russia
[2] Nikolaev Institute of Inorganic Chemistry SB RAS, 3 Acad. Lavrentiev Ave., 630090 Novosibirsk, Russia
[\*] Correspondence: avp-diamond@mail.ru

**Abstract:** Phase change and graphitization of diamonds from the Popigai impact crater (Krasnoyarsk Territory, Siberian platform, Russia) exposed to high-pressure high-temperature (HPHT) conditions of 5.5 GPa and 2000–2200 °C are studied by Raman spectroscopy and X-ray diffractometry (XRD). Light-color diamonds of type 1, free from inclusions, with 0 to 10 % lonsdaleite, are more resistant to HPHT effects than dark diamonds of type 2 rich in lonsdaleite and graphite. The lonsdaleite/diamond ratios in lonsdaleite-bearing impact diamonds become smaller upon annealing, possibly because lonsdaleite transforms to cubic diamond simultaneously with graphitization. Therefore, lonsdaleite is more likely a structure defect in diamond than a separate hexagonal phase.

**Keywords:** impact diamond; lonsdaleite; graphitization; high-pressure high-temperature (HPHT) experiment; XRD; Raman spectroscopy; treated sample

## 1. Introduction

The Popigai impact crater is located in the northern Siberian craton at the boundary between the Krasnoyarsk region and Yakutia. The impact origin of the crater was proven in 1971 by Victor L. Masaitis. The Popigai impact crater is a globally unique impact crater that stores inexhaustible resources of diamond with exceptional properties and is of great scientific and production interest. Many impact diamonds bear a lonsdaleite phase, are twinned, and have structural defects that divide crystals into subnanometer domains [1]. The Popigai lonsdaleite-bearing diamonds are paramorphs formed by shock-induced martensitic transformation of graphite in target gneiss into diamond as a result of a bolid impact on the Earth's surface [2]. The cited studies [3–11] have provided a wealth of data on macro- and micromorphology of diamond crystals, structure, isotope systematics, major- and trace-element chemistry, and petrology of the host rocks, as well as on the local geology of the crater site; transmission electron microscopy (TEM) revealed details of the diamond structure, with frequent twinning of nanometer crystals. Lonsdaleite was synthesized in numerous experiments, but no clear evidence that it is an independent phase has been obtained so far. Lonsdaleite has potentially hardness and rigidity exceeding those of cubic diamond [12], but these superior mechanical properties have never been proven experimentally due to the inability to synthesize lonsdaleite in a pure phase.

The Popigai impact diamonds are of several types, differing in relative percentages of cubic diamond, lonsdaleite, and graphite, which vary from sample to sample: colorless transparent crystals with 0 to 10% lonsdaleite (type 1), dark varieties consisting of cubic diamond, 40%–55% lonsdaleite, and graphite (type 2), and crystals of intermediate (3/2) type.

Numerous studies of natural impact diamonds provide a lot of interesting and useful material for understanding an impact event. Experimental studies on the synthesis of lonsdaleite-containing diamonds are also very interesting. At the same time, experiments on

annealing natural samples are not enough. Such experiments are limited by the complexity of working with such samples at high pressures and temperatures and the availability of such samples, in particular, diamonds from the Popigai astrobleme.

We investigated phase changes and graphitization in the Popigai impact diamonds exposed to high temperatures at 5.5 GPa for 180 s to 1200 s. Following [1], lonsdaleite is considered here as a faulted and twinned cubic diamond.

## 2. Materials and Methods

The experiments were performed in a split-sphere multi-anvil high-pressure apparatus [13–15], and methodic work was done on state assignment of IGM SB RAS. The apparatus does not have a press; its body consists of two opening semi-blocks, which are enfolded by two flange-type semi-cases. When closed, a spherical chamber is formed within the semi-blocks and is the space for a multi-anvil spherical guideblock. Two elastic membranes installed inside the semi-blocks separate the guideblock from the apparatus body. Both semi-blocks have channels for pumping oil under the membranes. The loading pressure is transmitted through the membranes to the guideblock. The multi-anvil spherical guideblock named "8/6" consists of, first, an outer (8) and, second, an inner (6) stage. The outer stage is a sphere with a diameter of 300 mm consisting of eight separate segments—steel anvils. The top of each anvil is truncated in the form of an equilateral triangle. Compressible plastic gaskets are installed between all anvils of the stage. As assembled, the split-sphere has an octahedral-shape chamber in its center designed to install six tungsten carbide anvils (WC6). Their truncated tops in turn form a parallelepiped-shape chamber within that is the space for a high-pressure cell. Pressure in the cell increases as a result of multiplication of load applied to the spherical outer block and is proportional to the surface ratio between the outer block and the truncated tops of WC anvils. Cold water flows through the inner-stage anvils and provides their efficient cooling.

The high-pressure cell had a parallelepiped shape with truncated edges, 23.0 × 20.5 mm in size, and was composed of compressed-powder refractory oxides $ZrO_2$ and CaO. The assembly included a tube graphite heater (0.5 mm thick walls, 10.0 mm inner diameter) placed in the cell center, with graphite and molybdenum discs on the top and at the base used as electrodes. Temperature and pressure were increased at rates of ~200 °C/min and 0.1–0.2 GPa/min, respectively. The temperature was monitored using a $PtRh_6/PtRh_{30}$ thermocouple till 1800 °C, and higher temperatures were estimated from the empirical temperature dependence of electric current plotted according to melting points of reference metals at a pressure of 5.5 GPa. Pressure was estimated using its empirical dependence on oil pressure in the hydraulic system and calibrated by recording changes in the resistance of PbSe, Ba, and Bi. The data were corrected for compression while heating with reference to the graphite-diamond equilibrium at 5 GPa and 1400 °C [16]. The pressure measurements were accurate to ±0.2 GPa. The reaction capsules were configured in different ways depending on the run conditions (Figure 1).

The experiments were applied to plate-shaped fragments of impact diamonds (with graphite paramorph) extracted from crushed bedrock from the Skalnoye deposit of the north-east of the Krasnoyarsk Territory, Siberian platform, Russia (the eastern part of the crater is located in the territory of the Sakha Republic, Russia). The crater diameter is approximately 100 km. Colorless and yellowish transparent diamonds of types 1, 3/2 and 2 and milky-white diamonds of type 1 (the last is free from lonsdaleite that is described by [17]) were selected (Figures 2 and 3).

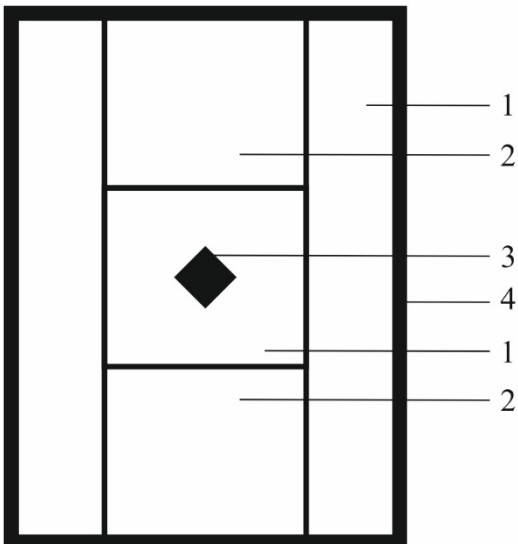

**Figure 1.** Reaction capsule with MgO powder (1), $ZrO_2$ and CsCl pellets (2), and diamond sample (3) in graphite heater (4).

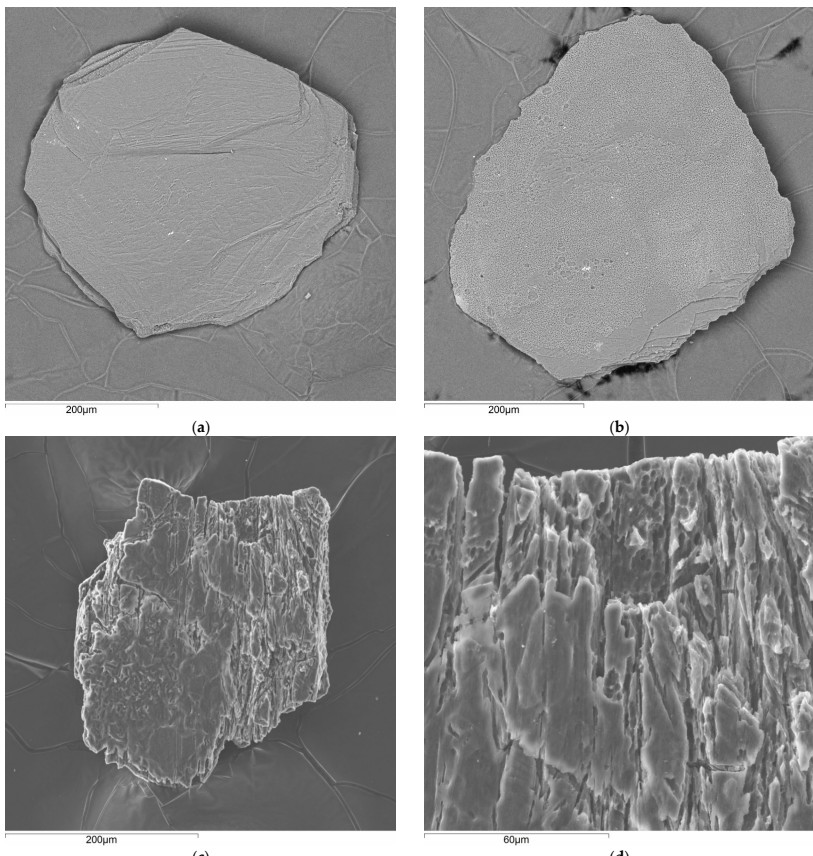

**Figure 2.** Representative Popigai impact diamond samples of types 1 (**a**), 3/2 (**b**), and 2 (**c**), and an enlarged fragment of type 2 diamond (**d**).

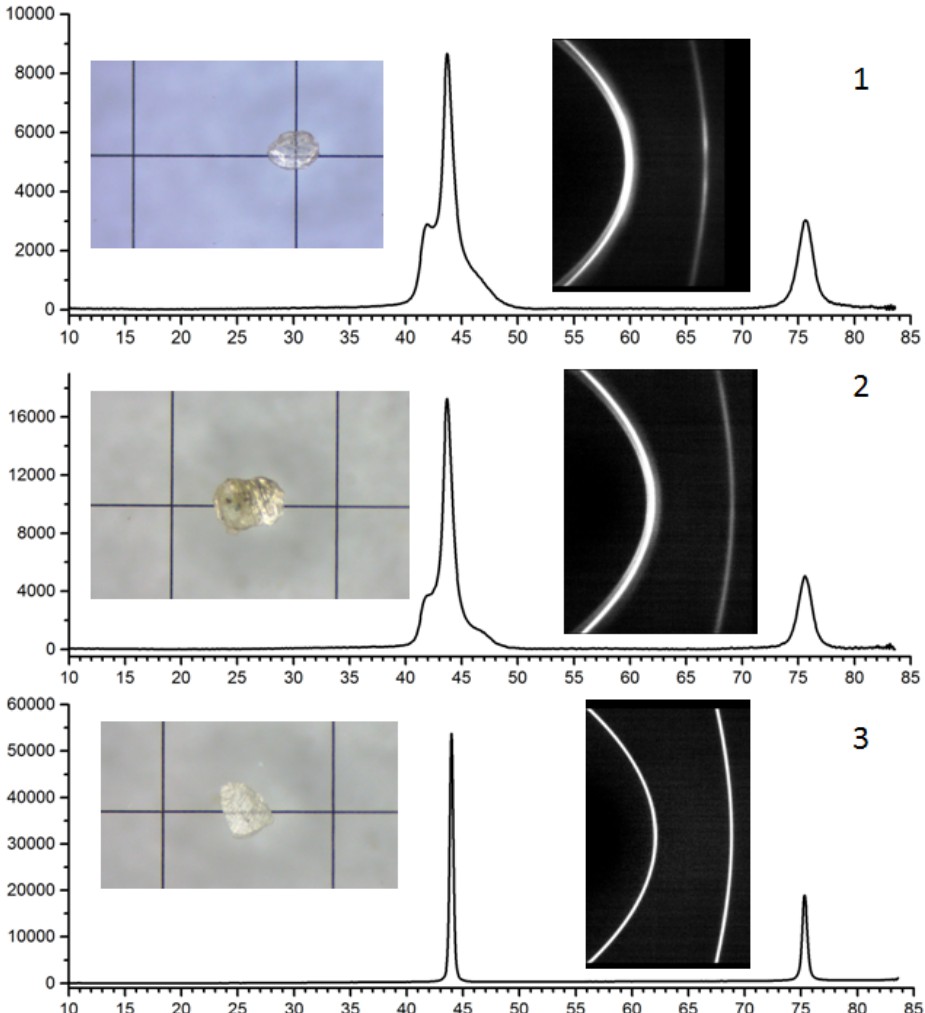

**Figure 3.** XRD data (Bruker D8 Venture diffractometer, CuKα-radiation) for colorless transparent diamonds of types 1 and 3/2 (**1**); yellowish transparent diamonds of type 3/2 (**2**), and milky-white diamonds of type 1, which is free from lonsdaleite (**3**).

The samples were compressed in MgO powder and placed in the middle of the reaction capsule between pellets from the compressed mixture powder $ZrO_2$ and CsCl, and the whole assembly was set inside the graphite heater.

The assembly was taken out of the cell and dismounted; the diamond samples were extracted from the MgO powder after runs. The samples were split into 50–100 μm particles. Some samples were soaked in a mixture of acids to remove surface graphite while the other samples were boiled in water with the aim of preserving the graphite. Thus, further Raman spectroscopy and XRD analysis were applied to different particles of the initial samples.

The Popigai diamond samples were 0.1–0.3 mm light-color transparent plates (type 1) or 0.1–0.3 mm dark plates containing more lonsdaleite and graphite (type 2), with their plate habits inherited from graphite in the target gneiss. The diamond samples were analyzed before and after the HPHT experimental runs by optical and scanning electron microscopy, Raman spectroscopy, and X-ray diffractometry at the Analytical Center of the V.S. Sobolev Institute of Geology and Mineralogy (Novosibirsk). The instruments were, respectively, an Olympus BX35 optical microscope, a MIRA LMU SEM microscope, and a JXA-8100 microanalyzer.

XRD analysis of impact diamond particles (Figure 3) was performed on a Bruker D8 Venture diffractometer (CuKα-and-Mo radiation, Incoatec IμS 3.0 microfocus tube, three-circle goniometer, PHOTON III CPAD detector, resolution 768 × 1024, pixel size

$135 \times 135$ μm$^2$) at 300 K. Diffraction curves were obtained as in [18] to completely solve the texture problem. During the XRD experiment, 20 Debye patterns were obtained for each particle at maximum different orientations of the sample relative to the primary beam. The diffraction arcs in the Debye patterns after summation (see insets in Figure 3) were almost perfectly filled, which ensured accurate intensity ratios of diamond to lonsdaleite defects (Lds). About 30 particles of different colors were investigated in this way, and several particles were selected for HPHT experiments and further analyses.

Raman spectra were excited with a UV line 325-nm He-Cd laser or a 532-nm neodymium laser (Nd:Y3Al5O12) line at an emission power of 5 mW per sample and were recorded on a Horiba Jobin Yvon LabRam HR800 spectrometer with a 1024-multichannel CCD detector (Andor) [8,19]. The light scattered by the sample in 180° geometry was collected on an Olympus BX41 microscope. The 3.5 cm$^{-1}$ spectral resolution of the recorded Stokes bands, at a Raman shift of 1330 cm$^{-1}$, was achieved using a holographic grating with 2400 grooves/mm, an equal 150 μm slit and a confocal hole diameter. Raman spectra were recorded from $-10$ to 3800 cm$^{-1}$. Wavenumbers were calibrated using neon lamp lines; the estimates of peak wavenumbers were accurate to $\pm 1$ cm$^{-1}$. The Raman spectra were deconvolved into Voigt amplitude contours using the PeakFit software package [20].

The content of lonsdaleite in initial and HPHT-treated diamond-lonsdaleite samples was determined from the spectral parameters according to the conventional procedure [7,8,21]. The calibration relationship was based on correlation between the lonsdaleite content and the width of the first-order total asymmetric Raman band characterizing the diamond-lonsdaleite (DL) mixture. The contributions of several L1 + Diam + L3 bands added up to this main band, as seen in figures 4 and 5 [8]. The lonsdaleite (*L*) percentages can be reliably estimated in this case, as it was shown by Raman microprobe analysis. Relative error in determination of lonsdaleite content is estimated as $(\Delta L)/L = \pm 0.07$ according to [8]. The Raman spectra of impact diamond contain graphite bands (G) at 1581 cm$^{-1}$ and disordered graphite bands (D1) at ~1420 cm$^{-1}$ (at 325-nm laser excitation).

## 3. Results

The Popigai impact diamonds exposed to high-pressure high-temperature (HPHT) conditions for 180 to 1200 s were studied with Raman microprobe spectroscopy. Below we report low-luminescence Raman spectra excited with a 325-nm UV laser and typical Raman scattering bands of impact diamond and graphite, but leave beyond consideration highly luminescent 532-nm spectra.

The percentages of lonsdaleite estimated from the lonsdaleite/diamond Raman intensity ratio as in [8,21] were 0 to 10% and 40 to 55% in diamond samples of types 1 and 2, respectively, and 20 to 30% in those of intermediate type 3/2. The Raman spectra of type 1 diamond before HPHT runs (Figure 4) included the first-order strongest band was in the 1325 to 1333 cm$^{-1}$ range. The spectra measured at three or four points proved sufficient homogeneity of the samples, with minor variations of diamond percentage appearing as a small shift within 2 cm$^{-1}$.

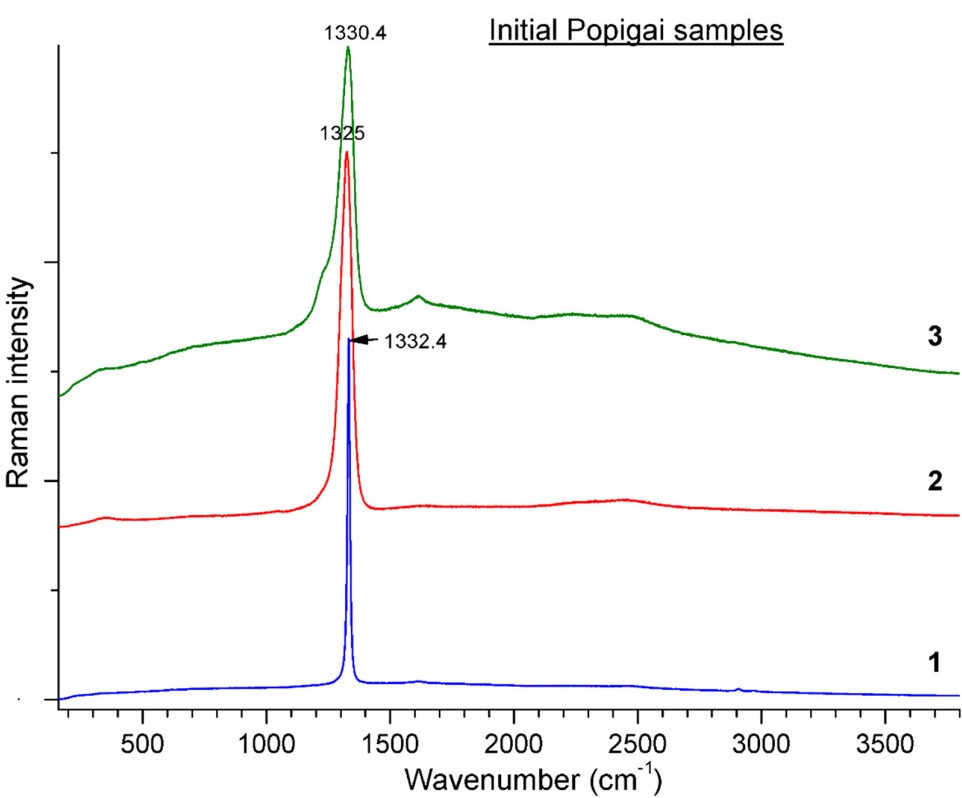

**Figure 4.** Raman spectra of representative impact diamonds of types 1 and 3/2: sample 3 (type 3/2) contains more lonsdaleite than sample 2 (type 1). Sample 1 is almost pure cubic diamond free from lonsdaleite (limiting variety of type 1 sample).

The second-order Raman spectrum of diamonds associated with density peaks of the two-phonon states lies in the 2100–2700 cm$^{-1}$ region and has a broad peak around ~2474 cm$^{-1}$. The ~330 cm$^{-1}$ low-frequency Raman band is presumably assigned to 'onion' structures with $sp^3$-packing of C atoms.

Deconvolution of the first-order Raman spectrum into constituent contours [7,8] revealed several main bands corresponding to cubic diamond (Diam.) at 1328–1333 cm$^{-1}$ and three lonsdaleite bands of different intensities at 1292 to 1307 cm$^{-1}$ (the strongest L1, $A_{1g}$ symmetry) [7,8,22]; 1235–1245 cm$^{-1}$ (less intense L2, $E_{2g}$); and presumably in the ~1337–1356 cm$^{-1}$ range (L3, $E_{1g}$) [8,19,23]. L2 shows up as a shoulder band of the first-order main Raman band, and L3 apparently overlaps the cubic diamond (Diam.) band. The spectrum of the type 2 diamond sample (Figure 5) revealed 42.7% lonsdaleite and a distinct amount of graphite. The Raman spectroscopy data are supported by XRD results for the diamonds of types 1 and 3/2 before the HPHT treatment (Figure 3). The diffraction intensity ratio of diamond and satellite lines in Figure 3 shows that lonsdaleite defects decrease in the sample series (1)-(2)-(3), with a particle (3) that is almost free from lonsdaleite (Figure 3).

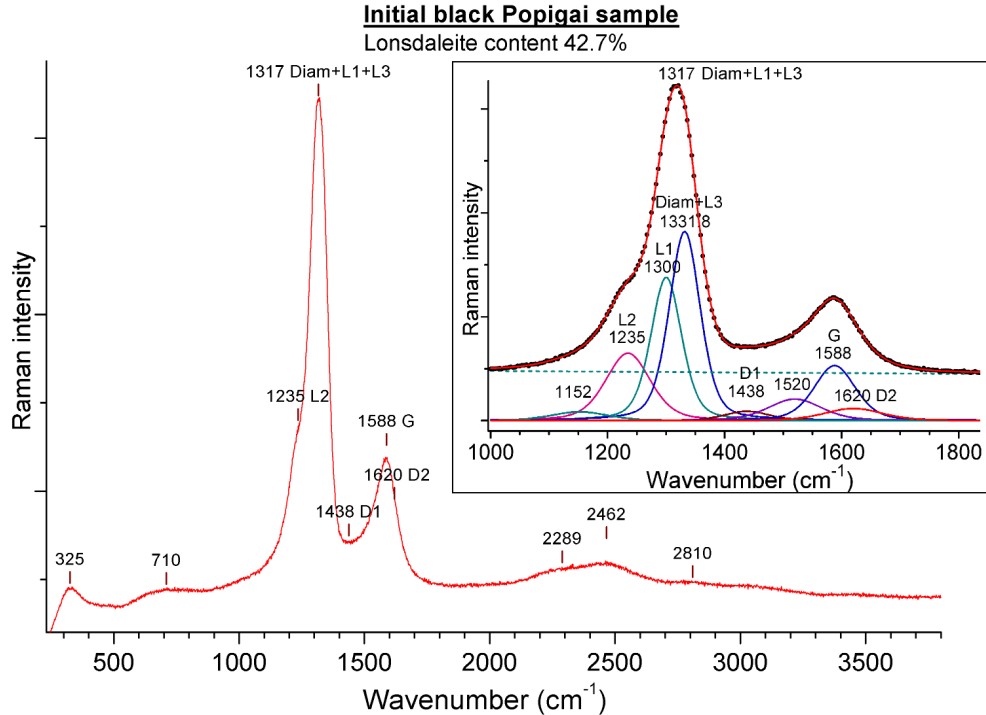

**Figure 5.** Raman spectrum of representative type 2 impact diamond (dark) sample with 42.7% lonsdaleite before HPHT runs. Inset shows first-order Raman spectrum deconvolved into Voigt contours.

The lonsdaleite percentage changes in diamonds exposed to high temperatures at 5.5 GPa, according to Raman spectroscopy, are summarized in Table 1 described in detail below.

**Table 1.** Changes in lonsdaleite percentages in Popigai impact diamonds after HPHT runs, according to Raman spectroscopy.

| Sample | Run | Temperature, T °C | Run Duration τ, s | Lonsdaleite Percentage, % |
|---|---|---|---|---|
| 1 | 4-12-22(7) | 2000 | 600 | 3.2 → 3.6–5.3; average 4.5 |
| 2 | 4-12-22(5) | 2050 | 600 | 8.0 → average 6.0 |
| 3 | 4-35-21 | 2100 | 180 | 45 → 7.6–6.3: average 7.1 |
| 4 | 4-46-21 | 2100 | 600 | 52 → 2.8–4.5: average 3.3 |
| 5 | 4-3-22(1) | 2050 | 1200 | 29.7 → 4.6–13.6; average 9.3 |
| 6 | 4-3-22(2) | 2100 | 1200 | 25.9 → 4.3–8.6; average 6.1 |

Note: Samples are light-color of type 1 (1, 2), dark of type 2 (3, 4), and light-color of intermediate type 3/2 (5, 6).

### 3.1. Graphitization of Impact Diamond

After the impact diamond samples were exposed to temperatures of 2000 °C or higher, graphite crystallized on their surfaces as druse-like aggregates of micrometer crystals.

#### 3.1.1. Samples of Type 1, with 0 to 10% Lonsdaleite (Runs 4-12-22-7 and 4-12-22-5)

Samples 1 and 2 of type 1 dissolved in the mixture of acids showed surface graphitization but preserved diamond cores, and the graphite layer became thicker as the temperature and run duration increased. The Raman spectra of the treated samples revealed graphite of well-crystallized and disordered modifications. The amount of well-crystallized graphite (G-band at 1590.5 cm$^{-1}$) was notably larger after than before the HPHT runs (Table 2, Figure 6).

**Table 2.** Raman spectra of sample 1 before (Ch2022-7b) and after (Ch2022-7a) HPHT run 4-12-22(7) at 5.5 GPa, 2000 °C, 600 s.

| Scheme | Lonsd | $\nu_{dia}$ | $I_{dia}$ | $w_{dia}$ | $w_{dia} \cdot I_{dia}$ | $\nu_g$ | $I_g$ | $w_g$ | $w_g \cdot I_g$ | $R_{dg}$ (Peak) | $R_{dg}$ | $R_{gd}$ |
|---|---|---|---|---|---|---|---|---|---|---|---|---|
| | % | cm$^{-1}$ | | cm$^{-1}$ | cm$^{-1}$ | cm$^{-1}$ | | cm$^{-1}$ | cm$^{-1}$ | | | |
| Ch7b-init2 | 3.6 | 1332.7 | 3400 | 19.4 | 65,960 | 1590.5 | 632 | 37.8 | 23,889.6 | 5.38 | 2.76 | 0.36 |
| Ch7a-1tr1 | 5.3 | 1332.3 | 1647 | 25 | 41,175 | 1588.2 | 454 | 44.5 | 20,203 | 3.63 | 2.038 | 0.49 |
| Ch7a-2tr1 | 3.6 | 1332.1 | 1572 | 19.4 | 30,497 | 1589.9 | 471 | 52 | 24,492 | 3.34 | 1.245 | 0.8 |
| Ch7a-3tr1 | 3.8 | 1331.7 | 522 | 20.4 | 10,649 | 1586.3 | 990 | 37.3 | 36,927 | 0.527 | 0.288 | 3.47 |
| Ch7a-3tr2 | 4.8 | 1332.4 | 488 | 23.4 | 11,419 | 1587.1 | 685 | 36.8 | 25,208 | 0.712 | 0.453 | 2.21 |

Note: Lonsd = Lonsdaleite percentage; $\nu$ = band wavenumber; $w$ = bandwidth; $I$ = peak intensity; $w \cdot I$ = integral intensity of total diamond-lonsdaleite (dia. or DL) and graphite (g) bands; $R_{dg} = I_{dia}/I_g$ is diamond/graphite ratio of integral intensity (and reverse $R_{gd} = 1/R_{dg}$). Total diamond-lonsdaleite band (dia) includes Diam. + L1 + L3 bands (Figure 5).

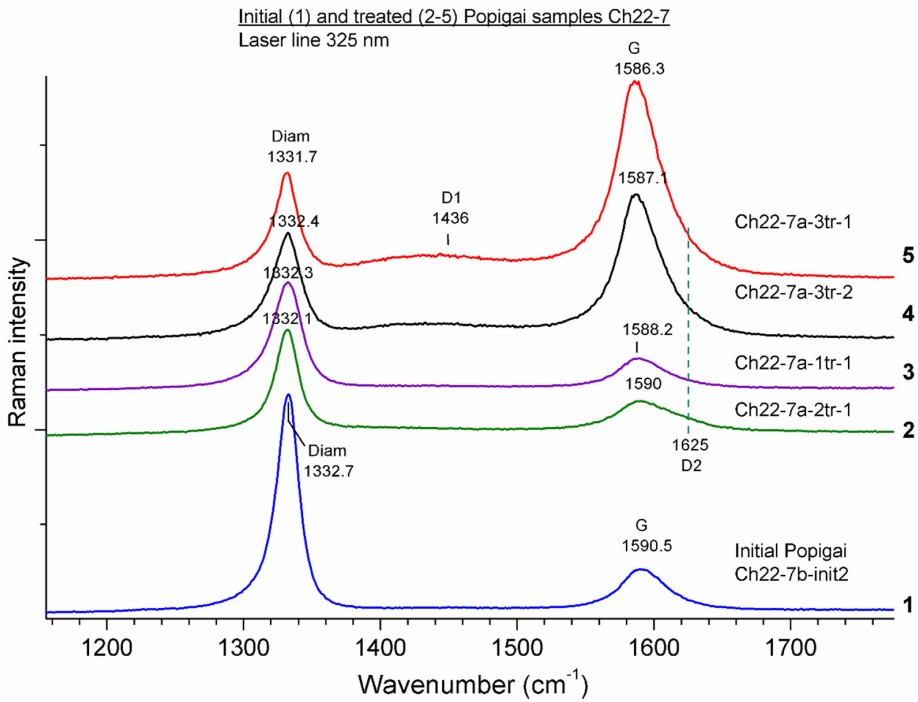

**Figure 6.** Raman spectra of type-1 impact diamond sample 1 and its fragments (Figure 2) before (Ch22-7b-init2, curve 1) and after (Ch22-7a-tr, curves 2–5) HPHT run 4-12-22(7) at 5.5 GPa and 2000 °C, 600 s.

Here and in figures below: impact diamond samples in compressed MgO powder were placed in the middle of the assembly between compressed $ZrO_2$ and CsCl inside a graphite heater. After the run, the spectra of diamonds in the region 1160 to 1780 cm$^{-1}$ were recorded in different sections under 325 nm UV laser excitation.

Less ordered graphite (Raman bands D1 1368 and G 1610 cm$^{-1}$) is present in minor amounts. Thus, diamond converted to graphite on the surface while the share of ordered graphite increased.

The graphite content increased significantly also in sample 2 of type 1 diamond (Ch2022-5-treat) exposed to high P-T conditions (Table 3, Figure 7): the relative intensity of the graphite band became 33 times greater than that of diamond (Figure 7, Table 3). The annealed sample showed a prominent graphite signal, unlike the initial sample with only almost invisible traces of graphite (Figure 7), while the graphite wavenumber de-

creased from 1609 to ~1589 cm$^{-1}$ (Table 3), indicating that diamond converted to better ordered graphite.

**Table 3.** Raman spectra of sample 2 before (Ch2022-5init) and after (Ch2022-5tr) HPHT run 4-12-22(5) at 5.5 GPa, 2050 °C, 600 s.

| Sample/Point | Lonsd | $\nu_{dia}$ | $I_{dia}$ | $w_{dia}$ | $\nu_{L2}$ | $\nu_g$ | $I_g$ | $w_g$ | $R_{dg}$ (peak) | $R_{dg}$ | $R_{gd}$ |
|---|---|---|---|---|---|---|---|---|---|---|---|
| | % | cm$^{-1}$ | | cm$^{-1}$ | cm$^{-1}$ | cm$^{-1}$ | | cm$^{-1}$ | | | |
| Ch-5init-1 | 8 | 1331.7 | 11,262 | 32.9 | 1238 | 1609 | 20 | 46 | 563 | 408 | 0.0024 |
| Ch-5tr-2 | 6 | 1331.6 | 11,282 | 33.6 | 1232 | 1588.7 | 651 | 46.6 | 17.3 | 12.5 | 0.08 |

Note: Lonsd = Lonsdaleite percentage; $\nu$ = band wavenumber; $w$ = bandwidth; $I$ = peak intensity; peak intensity ratio $R_{dg}$(peak); $R_{dg} = I_{dia}/I_g$ is diamond/graphite ratio of integral intensities (and reverse $R_{gd} = 1/R_{dg}$). Total diamond-lonsdaleite band (dia) includes Diam. + L1 + L3 bands.

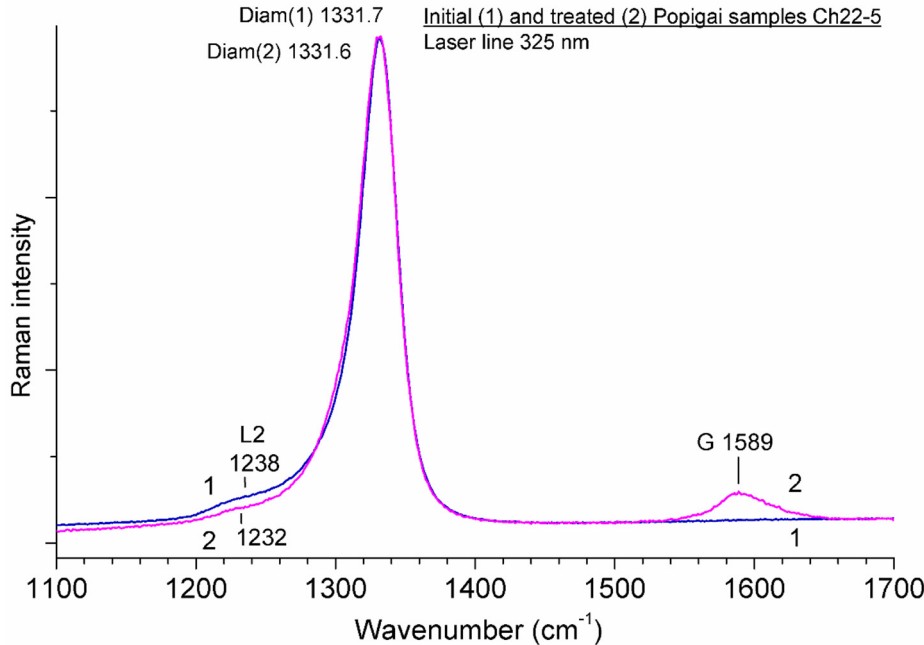

**Figure 7.** Raman spectra of sample 2 before (Ch2022-5init, curve 1) and after (Ch2022-5tr, curve 2) HPHT run 4-12-22(5) at 5.5 GPa, 2050 °C, 600 s.

### 3.1.2. Dark (Type 2) Diamond Samples 3 and 4 (40 to 55 mol.% Lonsdaleite), Runs 4-35-21 and 4-46-21

The Raman spectra of dark-colored type 2 diamonds contain a pronounced G band of graphite inclusions which impart the dark coloration to the diamond. The Raman spectra of samples 3 (Figure 8) and 4 (Figure 9) after HPHT runs 4-35-21 and 4-46-21, respectively, reveal two main graphite modifications, or their combination in some cases (spectrum 6 in Figure 8): crystallized graphite (strong narrow G band at 1582 cm$^{-1}$ and weak D1 band at 1427 cm$^{-1}$) and disordered amorphous graphite (strong broad G band at 1610 cm$^{-1}$ and weak broad D1 band at 1368 cm$^{-1}$). The second-order Raman spectrum of crystalline graphite is associated with two-phon density peaks in the region of 2700–3000 cm$^{-1}$, with broad bands at ~2328 and 2875 cm$^{-1}$ (2D1). Well-crystallized graphite appears as G (1583 cm$^{-1}$) and D1 (1350 cm$^{-1}$) bands, as well as 1620, 2464, and 2702 cm$^{-1}$ bands [21].

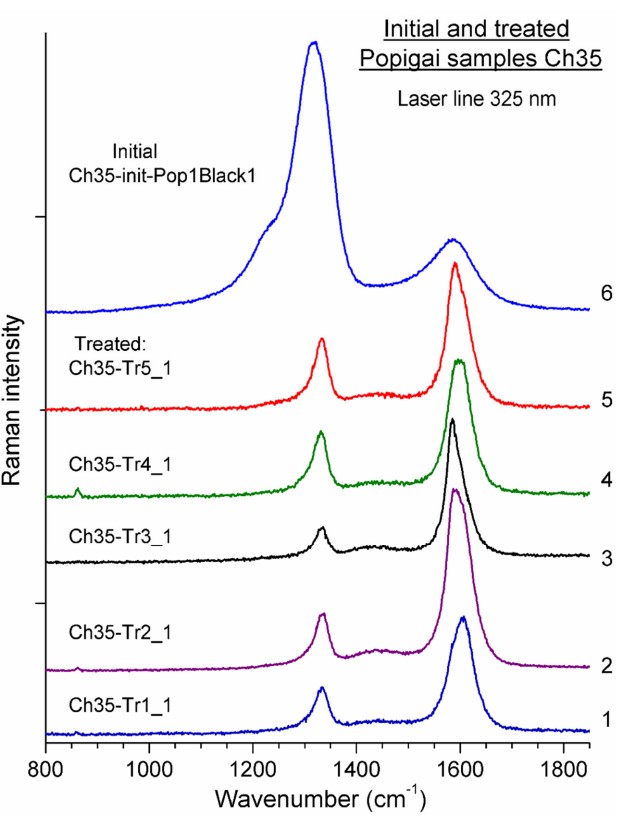

**Figure 8.** Raman spectra of sample 3 before (Ch35-init-Pop-1Black1, curve 6) and after (Ch35-Tr, curves 1–5) HPHT run 4-35-21 at 5.5 GPa, 2100 °C, 180 s.

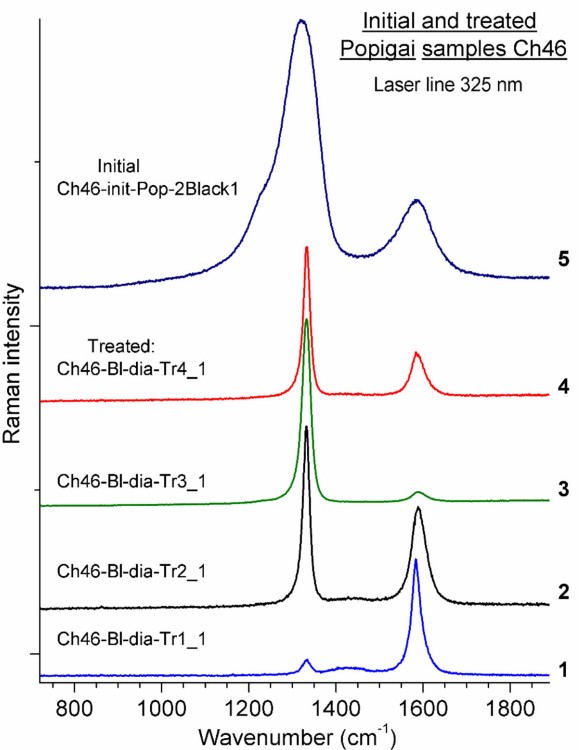

**Figure 9.** Raman spectra of sample 4 before (Ch46-init-Pop-2Black1, curve 5) and after (Ch46-Tr, curves 1–4) HPHT run 4-46-21 at 5.5 GPa, 2100 °C, 600 s.

Graphite in annealed sample 3 (Figure 8) likewise increased notably in amount and appeared in two modifications: nanostructured (G-band at ~1603 cm$^{-1}$) and better crystallized (G-band at ~1580–1590 cm$^{-1}$) graphite. At the same time, the fraction of cubic diamond in the diamond-graphite mixture decreased significantly, which shows up in a lower diamond/graphite Raman intensity ratio: ~0.2 against 3.8 (Table 4). At HPHT treatment, the wavenumber of total diamond-lonsdaleite band $\nu_{dia}$ is increased, which proves the disorder decrease and the enlargement of crystallites. The crystallite size controls the band broadening and low-frequency shift of the band according to the relations [24].

**Table 4.** Raman spectra of sample 3 before (Ch35-init-Pop-1Black1) and after (Ch35) HPHT run 4-35-21 at 5.5 GPa, 2100 °C, 180 s.

| Sample/ Grain/Point | Lonsd. | Intensity $I_{dia}/I_g$ | $\nu_{dia}$ | $w_{dia}$ | $\nu_{G1}$ | $\nu_{G2}$ | $w_{G12}$ | $\nu_{D12}$ | Comment |
|---|---|---|---|---|---|---|---|---|---|
| | % | Dimensionless | cm$^{-1}$ | cm$^{-1}$ | cm$^{-1}$ | cm$^{-1}$ | cm$^{-1}$ | cm$^{-1}$ | |
| Ch35-init-Pop-1Black1 | 45 | 3.8 | 1317.2 | 82.5 | 1588 | - | 93.6 | - | min $\nu_{dia}$, high lonsd. content |
| Ch35-Tr1-1 | 7.63 | 0.22 | 1333.6 | 31.9 | 1588.8 | 1606 | 53.5 | 1435 | |
| Ch35-Tr1-2 | 6.9 | 0.24 | 1333.0 | 29.7 | 1590 | 1603 | 50.8 | 1435 | |
| Ch35-Tr2-1 | 7.3 | 0.16 | 1334.0 | 31.0 | 1590 | 1602 | 53.2 | 1439 | |
| Ch35-Tr2-2 | 6.5 | 0.16 | 1333.9 | 28.7 | 1589.7 | 1605 | 47.8 | 1437 | |
| Ch35-Tr3-1 | 6.3 | 0.15 | 1334.0 | 28.1 | 1584.9 | ~1600 | 38.5 | 1426 | |
| Ch35-Tr4-1 | 7.6 | 0.26 | 1332.0 | 31.7 | 1593 | 1605 | 52.3 | 1440 | |
| Ch35-Tr4-2 | 7.2 | 0.2 | 1335.8 | 30.7 | 1588 | 1600 | 43.2 | 1434 | max $\nu_{dia}$ |
| Ch35-Tr5-1 | 7.55 | 0.3 | 1333.6 | 31.7 | 1590.7 | 1606 | 47.7 | 1444 | |

Note: Lonsd = Lonsdaleite percentage; $\nu$ = band wavenumber; $w$ = FWHM bandwidth; diamond/graphite ratio of integral intensities $I_{dia}/I_g$. Total diamond-lonsdaleite band (dia. or DL) includes overlapped Diam + L1 + L3 bands. $w_{G12}$ is total width of overlapped G1 and G2 graphite bands. The D12 hump is two overlapped bands of disordered G1 graphite and G2 graphite: D1(1) and D1(2), respectively. Kimberlitic diamond band: $\nu_{dia}$ = 1332.3 cm$^{-1}$.

The amount of graphite in different grains of sample 4 exposed to HPHT conditions in run 4-46-21 increased generally but showed large variance (Figure 9; Table 5). The Raman spectra likewise revealed graphite of two modifications: a G2-band at ~1603 cm$^{-1}$ corresponding to nanoscale graphite and a G1-band at ~1580–1590 cm$^{-1}$ of well-crystallized graphite. The G2 band at ~1603 cm$^{-1}$ appears as a low-intensity shoulder band of G1 (~1580–1590 cm$^{-1}$). At the same time, the fraction of cubic diamond in the diamond-graphite mixture decreased significantly, likewise with large variance, expressed as lower diamond/graphite intensity ratio: ~0.1–0.2 against 3.9 (Table 5).

**Table 5.** Raman spectra of sample 4 before (init-Pop-2Black1) and after (Ch46) HPHT run 4-46-21 at 5.5 GPa, 2100 °C, 600 s.

| Sample-Grain-Point | Lonsd. | Intensity $I_{dia}/I_g$ | $\nu_{dia}$ | $w_{dia}$ | $\nu_{G1}$ | $\nu_{G2}$ | $w_{G12}$ | $\nu_{D12}$ | Comment |
|---|---|---|---|---|---|---|---|---|---|
| | % | Unitless | cm$^{-1}$ | cm$^{-1}$ | cm$^{-1}$ | cm$^{-1}$ | cm$^{-1}$ | cm$^{-1}$ | |
| Ch46-init-Pop-2Black1 | 52 | 3.9 | 1322.0 | 89.7 | 1586 | - | 84.3 | - | Highest lonsd.cont. |
| Ch46-Tr1-1 | 4.5 | 0.11 | 1332.3 | 22.4 | 1583.6 | - | 23.8 | 1426 | Narrow G1 |
| Ch46-Tr2-1 | 2.9 | 0.83 | 1332.1 | 17.4 | 1589.3 | 1608 | 39 | 1440 | |
| Ch46-Tr3-1 | 2.9 | 2.0 | 1332.3 | 17.2 | 1588.5 | 1609 | 38.8 | - | |
| Ch46-Tr4-1 | 2.8 | 1.68 | 1332.9 | 17.1 | 1586.8 | 1601 | 36.1 | 1429 | Max $\nu_{dia}$ |

Note: Lonsd = Lonsdaleite percentage; $\nu$ = band wavenumber; $w$ = FWHM bandwidth; diamond/graphite ratio of integral intensities $R_{dg} = (I_{dia}/I_g)$. Total diamond-lonsdaleite band (dia. or DL) includes overlapped Diam + L1 + L3 bands. $w_{G12}$ is total width of overlapped G1 and G2 graphite bands. The D12 hump is two overlapped bands of disordered G1 graphite and G2 graphite: D1(1) and D1(2), respectively. Kimberlitic diamond band: $\nu_{dia}$ = 1332.3 cm$^{-1}$.

### 3.2. XRD Analysis of Impact Diamond Samples of Types 1 and 2 before and after HPHT Runs

Graphitization of diamond samples exposed to 2000 °C was minor in particles of type 1 free from lonsdaleite (Figure 10a) but much greater in particles of type 2 (Figure 10b).

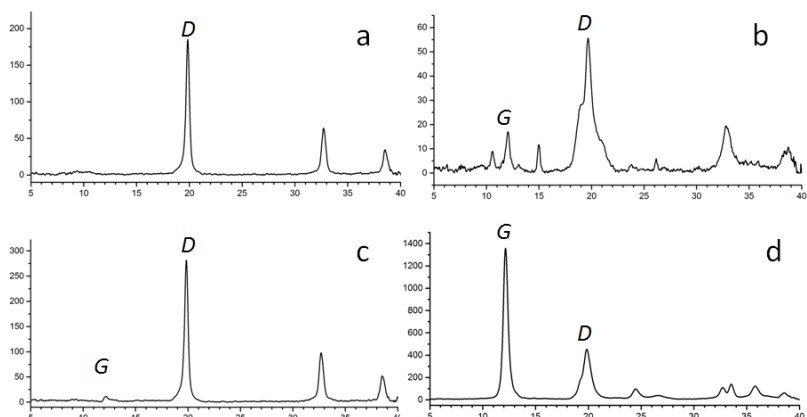

**Figure 10.** XRD analysis (Bruker D8 Venture diffractometer, MoKα-radiation) of impact diamonds. (**a**,**b**): samples of types 1 (**a**) and 2 (**b**) before HPHT runs; (**c**,**d**): samples of types 1 (**c**) and 2 (**d**) after HPHT runs at 5.5 GPa, 2000 °C, 180 s. D and G are the strongest reflections of diamond and graphite, respectively.

Diamond Samples 5 and 6 of Intermediate 3/2 Type (20 to 30 mol % Lonsdaleite), Runs 2022-1 and 2022-2)

The fraction of graphite in diamond of intermediate 3/2 type (sample 2022-1) exposed to HPHT conditions became much larger and the fraction of cubic diamond decreased correspondingly, which showed up in the change of the diamond/graphite integral intensity ratio $I_d/I_g$ from ~2.5 to ~0.4 (on average). The spectrum of the initial sample contains a complex graphite band at ~1583 and 1608 cm$^{-1}$ produced by two graphite modifications (Figure 11; Table 6). The magnitude of the graphite band width in treated samples varied from 40 to 57 cm$^{-1}$ in different particles and at different points within each particle, i.e., diamond conversion to graphite was heterogeneous. The newly formed less oxidized graphite with a G-band at ~1588–1595 cm$^{-1}$ showed a larger average bandwidth than that from the initial sample: ~49 cm$^{-1}$ against ~29 cm$^{-1}$.

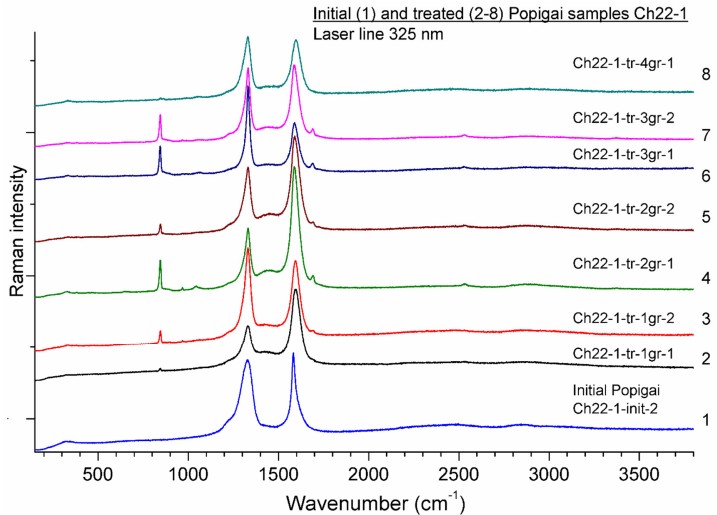

**Figure 11.** Raman spectra (150 to 3800 cm$^{-1}$) of dark sample 5 before (Ch22-1, curve 1) and after (Ch22-1-Tr, curves 2–5) HPHT run 4-3-22-1 at 5.5 GPa, 2050 °C, 1200 s. The curve labels refer to particle 1 at two points (2 and 3); particle 2 at two points (4 and 5); particle 3 at two points (6 and 7); and particle 4 at one point (8). In addition to the diamond-lonsdaleite and graphite bands, the spectra contain narrow resonant Raman peaks of $(CrO_4)^{2-}$ impurity at $\nu_0 = 845$ cm$^{-1}$, $2\nu_0 = 1693$ cm$^{-1}$ and $3\nu_0 = 2534$ cm$^{-1}$.

**Table 6.** Raman spectra of sample 5 before (Ch22-uv1init-2) and after (Ch22-uv-1tr) HPHT run 4-3-22-1 at 5 GPa, 2050 °C, 1200 s.

| Sample/Grain/ Point | Lonsd | Intensity $I_{dia}/I_g$ | $v_{dia}$ | $w_{dia}$ | $2v_{dia}$ | $v_{G1}$ | $v_{G2}, v_{D2}$ | $w_{G12}$ | $v_{D1}$ | $2v_{D1}$ |
|---|---|---|---|---|---|---|---|---|---|---|
| | % | Dimensionless | cm$^{-1}$ | cm$^{-1}$ | cm$^{-1}$ | cm$^{-1}$ | cm$^{-1}$ | cm$^{-1}$ | cm$^{-1}$ | cm$^{-1}$ |
| Ch22-uv1init-2 (initial Popigai) | 29.7 | 2.5 | 1329.1 | 73.2 | 2472 | 1583 | 1608, 1630 | 28.7 | 1425 | 2842 |
| Ch22-uv-1tr-1grain-1 | 12.7 | 0.6 | 1331 | 44.8 | 2525 | 1595.3 | - | 54.6 | 1427 | 2858 |
| Ch22-uv-1tr-1grain-2 | 10.4 | 0.89 | 1331.6 | 39.4 | 2484 | 1594.3 | - | 51 | 1424 | 2857 |
| Ch22-uv-1tr-2grain-1 | 8.3 | 0.33 | 1331.4 | 33.7 | 2314, 2480 | 1588.8 | - | 46.4 | 1436 | 2883 |
| Ch22-uv-1tr-2grain-2 | 10.7 | 0.88 | 1331.5 | 40.1 | 2480, 2530 | 1590 | 1605, 1626 | 52.8 | 1449 | 2865 |
| Ch22-uv-1tr-3grain-1 | 4.6 | 0.91 | 1331.2 | 22.9 | 2470, 2528 | 1589.2 | - | 46 | 1443 | 2868 |
| Ch22-uv-1tr-3grain-2 | 7.4 | 0.62 | 1331.8 | 31.2 | 2304, 2528 | 1587.8 | - | 47.3 | 1438 | 2875 |
| Ch22-uv-1tr-4grain-1 | 13.6 | 0.87 | 1330 | 47 | 2272, 2494 | 1597.4 | - | 56.7 | 1444 | 2873 |
| Ch22-uv-1tr-5grain-2 | 7.5 | 0.16 | 1331.4 | 31.6 | 2285, 2453 | 1587.9 | - | 40 | 1433 | 2886 |
| Ch22-uv-1tr-5grain-3 | 8.1 | 0.26 | 1332.1 | 33.1 | 2278, ~2528 | 1592.2 | - | 43.5 | 1428 | 2976 |

Note: Lonsd = Lonsdaleite percentage; $v$ = band wavenumber; $w$ = FWHM bandwidth; diamond/graphite ratio of integral intensities ($I_{dia}/I_g$). Total diamond-lonsdaleite band (dia. or DL) includes overlapped Diam + L1 + L3 bands. G12 is two overlapped G1 and G2 graphite bands. Kimberlitic diamond band: $v_{dia}$ = 1332.3 cm$^{-1}$. Additionally, narrow resonance peaks of $CrO_4^{2-}$ appear in the spectra of treated samples at $v_0$ = 845 cm$^{-1}$, $2v_0$ = 1693 cm$^{-1}$ and $3v_0$ = 2534 cm$^{-1}$ (Figure 12). $2v_{dia}$ and $2v_{D1}$ refer to the wavenumbers of second-order Raman peaks of diamond-lonsdaleite at ~2·$v_{dia}$ and graphite at ~2·$v_{D1}$.

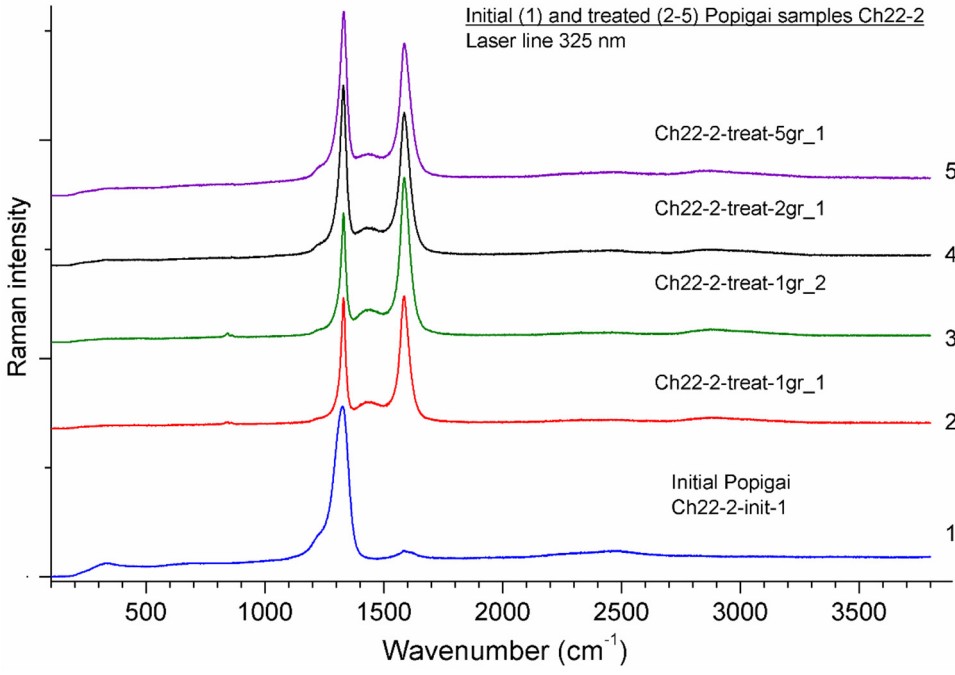

**Figure 12.** Raman spectra of sample 6 before (InitialPopigaCh22-2-init-1, curve 1) and after (Ch22-2-treat-1gr, curves 2–5) HPHT run 4-3-22-2 at 5.5 GPa, 2050 °C, for 1200 s. The curve labels refer to particle 1 at two points (2 and 3); particle 2 at one point (4); and particle 5 at one point (5). In addition to the diamond-lonsdaleite and graphite bands, the spectra contain a narrow resonant Raman peak of $(CrO_4)^{2-}$ impurity at $v_0$ = 845 cm$^{-1}$.

As in the case of sample 5, the fraction of graphite in diamond sample 6 of intermediate 3/2 type exposed to HPHT conditions in runs 2022-1 and 2022-2 became much larger and the fraction of cubic diamond decreased correspondingly, which showed up in the changed diamond/graphite integral intensity ratio $I_{\text{dia}}/I_{\text{g}}$ (Figure 12; Table 7). The graphite bandwidth in the treated sample likewise varied from 40 to 57 cm$^{-1}$ in different particles and at different points within each particle, i.e., diamond conversion to graphite was heterogeneous. However, the newly formed less oxidized graphite with a G-band at ~1588–1595 cm$^{-1}$ showed a smaller average bandwidth than that in the initial sample: ~46 cm$^{-1}$ against ~85 cm$^{-1}$.

**Table 7.** Raman spectra of sample 6 before (InitialPopigaCh22-2-init-1, curve 1) and after (Ch22-2-treat-1gr, curves 2–5) HPHT run 4-3-22-2 at 5.5 GPa, 2050 °C, for 1200 s.

| Sample/Grain/Point | Lonsd. | Intensity $I_{\text{dia}}/I_{\text{g}}$ | $\nu_{\text{dia}}$ | $w_{\text{dia}}$ | $2\nu_{\text{dia}}$ | $\nu_{\text{G1}}$ | $\nu_{\text{G2}}$ | $w_{\text{G1}}$ | $\nu_{\text{D1}}$ |
|---|---|---|---|---|---|---|---|---|---|
| | % | Dimensionless | cm$^{-1}$ | cm$^{-1}$ | cm$^{-1}$ | cm$^{-1}$ | cm$^{-1}$ | cm$^{-1}$ | cm$^{-1}$ |
| Ch22-uv-2i-1 (initial Popigai) | 25.9 | 16 | 1326.4 | 68.5 | 2474 | 1585.6 | no | 84.7 | - |
| Ch22-uv-2tr-1grain-1 | 4.3 | 0.5 | 1329.9 | 21.9 | 2473 | 1585 | no | 42.9 | 1436 |
| Ch22-uv-2tr-1grain-2 | 4.6 | 0.39 | 1330.1 | 22.7 | 2457 | 1586 | no | 44.3 | 1438 |
| Ch22-uv-2tr-2grain-1 | 6.9 | 0.76 | 1329.4 | 29.7 | 2458 | 1585.8 | no | 46.4 | 1432 |
| Ch22-uv-2tr-5grain-1 | 8.6 | 0.88 | 1331.1 | 34.5 | 2468 | 1586.6 | no | 48.3 | 1432 |

Note: Lonsd = Lonsdaleite percentage; $\nu$ = band wavenumber; $w$ = FWHM bandwidth; diamond/graphite ratio of integral intensities ($I_{\text{dia}}/I_{\text{g}}$). Total diamond-lonsdaleite band (dia. or DL) includes overlapped Diam + L1 + L3 bands. Kimberlitic diamond band: $\nu_{\text{dia}}$ = 1332.3 cm$^{-1}$.

In addition, spectra 1–5 (Figure 12) reveal silicate glass (SiO$_2$) coexisting with crystalline graphite (broad band of O-T-O bending vibrations in the range of 250–500 cm$^{-1}$) and air nitrogen (narrow band at 2331 cm$^{-1}$).

### 3.3. Diamond/Lonsdaleite Ratio in Raman Spectra of Impact Diamonds before and after HPHT Treatment

According to Raman spectroscopy, HPHT effects on the Popigai impact diamond samples caused notable changes to lonsdaleite percentages (Table 1).

3.3.1. Light-Color Type 1 Diamond Samples 1 and 2 (0 to 10% Lonsdaleite), Runs 4-12-22-5 and 4-12-22-7

The percentage of lonsdaleite in sample 1 exposed to HPHT conditions in run 4-12-22-7 did not differ much from that in the initial sample: 3.6%–5.3% against 3.6%, and was similar at 2–3 points measured in each particle (Figure 6; Table 2). The *L* value (Table 2) either did not change or slightly increased for lonsdaleite contents (Lonsd, %) estimated by bandwidths [21]) but slightly decreased according to estimates as in [8], based on band intensity ratio (including L2-band at 1244 cm$^{-1}$). Thus, the diamond/lonsdaleite ratio of type 1 diamond either changed slightly or rather remained invariable within the measurement accuracy. The bandwidth ($w$) slightly increased after exposure to high temperatures at 5.5 GPa, while the frequencies of diamond $\nu$ (dia) and graphite bands showed a minor decrease (Table 2).

Another diamond sample of type 1 (2) likewise retained a similar lonsdaleite percentage after HPHT run (4-12-22-5): 6% in Ch2022-5-5treat against 8% in Ch2022-5-initial (Figure 7; Table 3), and the values at 2–3 measured points were similar. The *L* value (Table 3) did not change in the estimates according to bandwidths [21]) but decreased from 8 to 6 % in the estimates based on band intensity ratio [8] (including L2-band 1244 cm$^{-1}$). Thus, the diamond/lonsdaleite ratio of type 1 diamond with 0 to 10 % lonsdaleite remained almost invariable (Table 3). The diamond band wavenumber (1331.7 cm$^{-1}$) and width changed only slightly within the measurement accuracy, while the intensity of the L2 band (1232–1238 cm$^{-1}$) decreased markedly (Figure 3).

To sum up, the lonsdaleite percentage in type 1 impact diamond exposed to HPHT conditions showed almost no change. The relative percentages of components in the dia-

mond/lonsdaleite mixture underwent minor or no change (lonsdaleite decreased slightly within the measurement accuracy).

### 3.3.2. Dark (Type 2) Samples 3 and 4 with 40 to 55% Lonsdaleite, Runs 4-35-21 and 4-46-21

The bands in the Raman spectra of type-2 diamond sample 3 became narrower after the HPHT run 4-35-21 (Figure 8): the width of the total diamond-lonsdaleite band decreased from 82.5 cm$^{-1}$ to 31 cm$^{-1}$ on average. The fraction of the cubic phase in the diamond-lonsdaleite mixture increased significantly, while the amount of lonsdaleite decreased from 45% to ~7% (on average).

The HPHT effects in sample 3 caused the diamond band to increase from 1317.2 cm$^{-1}$ to 1332–1335.8 cm$^{-1}$, till a maximum of 1335.8 cm$^{-1}$ at point 2 of particle 4 (Table 1). Note that this frequency exceeds that of the 1332.3 cm$^{-1}$ band for kimberlitic diamond in most of the particles (1–5) and points of the sample. Therefore, the structure of the treated sample became denser than in kimberlitic diamond.

The total diamond-lonsdaleite band in the spectra of sample 4 (Figure 9) reduced from 89.7 cm$^{-1}$ before to ~18 cm$^{-1}$ after the treatment and became narrower than in sample 3. The fraction of the cubic diamond phase in the diamond-lonsdaleite mixture of sample 4 increased up to almost 100%, while the share of lonsdaleite decreased from 52% to ~3% (Tables 1 and 5).

The frequency of the diamond band in sample 4 increased from 1322 cm$^{-1}$ before to 1332.1–1332.9 cm$^{-1}$ after the HPHT run (Table 4) and was the largest in particle 4 (1332.9 cm$^{-1}$, larger than 1332.3 cm$^{-1}$ in kimberlitic diamond). This increase observed in the Raman data can be explained by the formation of ultradense sheared diamond.

In addition to the diamond and graphite bands, the Raman spectra contained narrow resonant peaks of $CrO_4^{2-}$ at $\nu_0$ = 845 cm$^{-1}$ as the main mode and its harmonics: $2\nu_0$= 1693 cm$^{-1}$ and $3\nu_0$ = 2534 cm$^{-1}$ (Figures 1–5). The $CrO_4^{2-}$ impurities are traces of chromium-bearing acid, which was preserved in microcracks after graphite had been removed from the surface of HPHT-treated samples by leaching in acids and subsequent rinsing with water.

### 3.3.3. Samples 5 and 6 of Intermediate 3/2 Type (20 to 30 mol % Lonsdaleite), Runs 4-3-22-1 and 4-3-22-2

*Sample 5 (run 22-1)*

The share of cubic diamond in the diamond-lonsdaleite mixture of sample 5 increased significantly (Table 6). The bands in the Raman spectra of the treated samples (Figure 11) narrowed down from ~73.2 to 23–47 cm$^{-1}$ (35 cm$^{-1}$ on average). The large range from 23 to 47 cm$^{-1}$ in different points within a particle indicates heterogeneous lonsdaleite-diamond-graphite transformation, possibly because the exposure time was too short. The diamond band became slightly stronger after the treatment: ~1330.1–1332.1 cm$^{-1}$ instead of 1329.1 cm$^{-1}$ (Table 6).

*Sample 6 (run 22-2)*

The spectrum of sample 6 with 26% lonsdaleite contained a broad diamond-lonsdaleite band of 68.5 cm$^{-1}$ (Figure 12; Table 7). The share of cubic diamond in the diamond-lonsdaleite mixture in the treated sample likewise became much greater (Table 7), while the diamond-lonsdaleite band narrowed down. The diamond band ($\nu_{dia}$) broadened from 1326.4 cm$^{-1}$ before to ~1329.4–1332.1 cm$^{-1}$ after the HPHT run.

### 3.4. XRD Analysis of Type 2 Impact Diamond Samples before and after HPHT Runs

The diffraction pattern for a particle cleaned from surface graphite in a mixture of acids showed a LD decrease after the fragments of the particle were exposed to HPHT conditions (compare curve 1 with curves 2 (run 4-21-21) and 3 (run 4-18-21) in Figure 13).

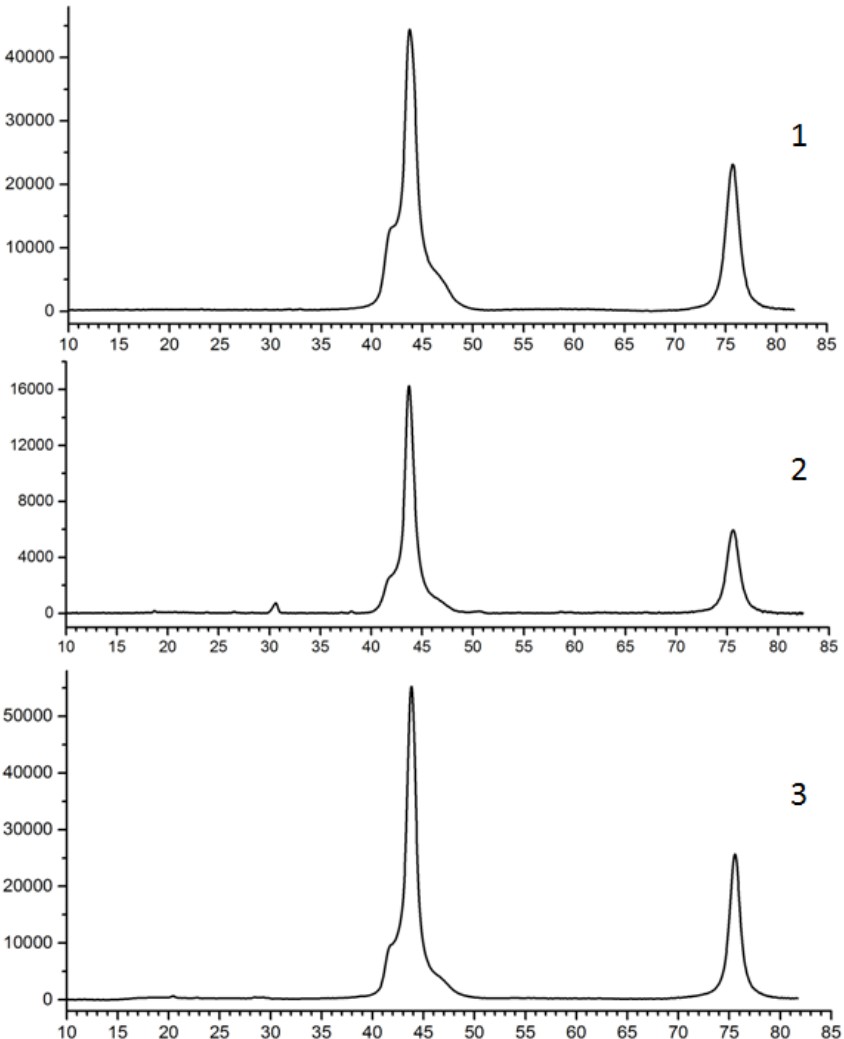

**Figure 13.** XRD analysis (Bruker D8 Venture diffractometer, CuKα-radiation) of type 3/2 impact diamond, before (1) and after (2) HPHT runs at 5.5 GPa, 1900 °C, for 60 s (run 4-18-21).

### 4. Discussion

The Popigai impact diamonds consist of nanometer crystals of diamond and lonsdaleite, with or without graphite: cubic diamond with a 3C structure (space group Fd3m) intergrown with 2H lonsdaleite (space group P6₃/mmc). Ultra-high-pressure experiments show that lonsdaleite can form as an independent phase in meteoritic bodies [3,4,25–27]. The hexagonal diamond has been synthesized under conditions of static pressure > 12 GPa and temperature > 1300 K [3]. It was found [7] that lonsdaleite might be obtained under pressures of 5.5–12 GPa with shear deformation and at temperatures of 1070–1600 K. Experiments on shock compression of pyrolytic and polycrystalline graphite at pressures from 19 GPa up to 228 GPa and ultrafast in situ X-ray diffraction measurements proved diamond formation on nanosecond timescales [25]. They record the direct formation of lonsdaleite above 170 GPa for pyrolytic samples only and state that lonsdaleite synthesis is highly difficult by applying static pressure but can be formed in violent impact events. Diamond and lonsdaleite are found to form together within bands with a core-shell structure following the high-pressure treatment of a glassy carbon precursor at room temperature above 80 GPa in diamond anvil cell, and it has been suggested that diamond forms from lonsdaleite during decompression [27].

The formation of lonsdaleite (wurzite-like phase P63/mmc, two-layer packing...ABAB) was attributed either to (i) corrugation or (ii) longitudinal bending of graphene layers. In the conditions of martensitic transformation, graphite converts into impact diamond

almost instantaneously, while lonsdaleite forms only at the expense of well-crystallized graphite. At the same time, lonsdaleite-bearing impact diamond inherits the morphology and structure of graphite particles, i.e., is a paramorph. Various theoretical models interpret disorder in impact diamonds in terms of (i) a diamond-lonsdaleite mixture, (ii) a disordered set of cubic and hexagonal layers along the [111] direction, and (iii) various disordering mechanisms (dislocations, stacking faults, etc.) described by stackograms [28].

On the other hand, there are solid proofs that, rather than being an independent phase, lonsdaleite is a defect structure of cubic diamond that forms on a microsecond timescale by solid-phase transformation of graphite under ultra-high pressure [1,5,29]. Lonsdaleite is metastable and lacks its own domain in the carbon phase diagram: there are no signatures of lonsdaleite formation under pressures and temperatures corresponding to the graphite/diamond equilibrium, while reverse diamond-to-lonsdaleite conversion is impossible. Lonsdaleite never occurs in nature as a separate phase but is always found intergrown with diamond. It was interpreted as stacking faults in the cubic diamond structure, a non-equilibrium product that forms by graphite explosion at ultra-high pressures and temperatures, simultaneously with cubic diamond [5]. The surface features of lonsdaleite-bearing diamonds coincide with twins and stacking faults of cubic diamond at the nanometer scale [1]. The diamond samples with diffraction patterns typical of lonsdaleite show intense twinning and structure defects that divide the crystals into sub-nanometer domains [1,30]. Furthermore, minor amounts of lonsdaleite are always present in any diamond as structure defects corresponding to a hexagonal lattice. This character of lonsdaleite in apographite martensitic diamond is confirmed by Raman spectroscopy: the spectra contain only diamond bands, though notably broadened.

As shown by Raman spectroscopy, lonsdaleite was preserved in impact diamonds for several hours (without spectra change), i.e., it was stable at 5.5 GPa and 1650 °C. Above 1800 °C and at the same pressure of 5.5 GPa, small flakes of well-crystallized graphite appeared in progressively increasing amounts on the surface of type 1 diamonds (Figure 14).

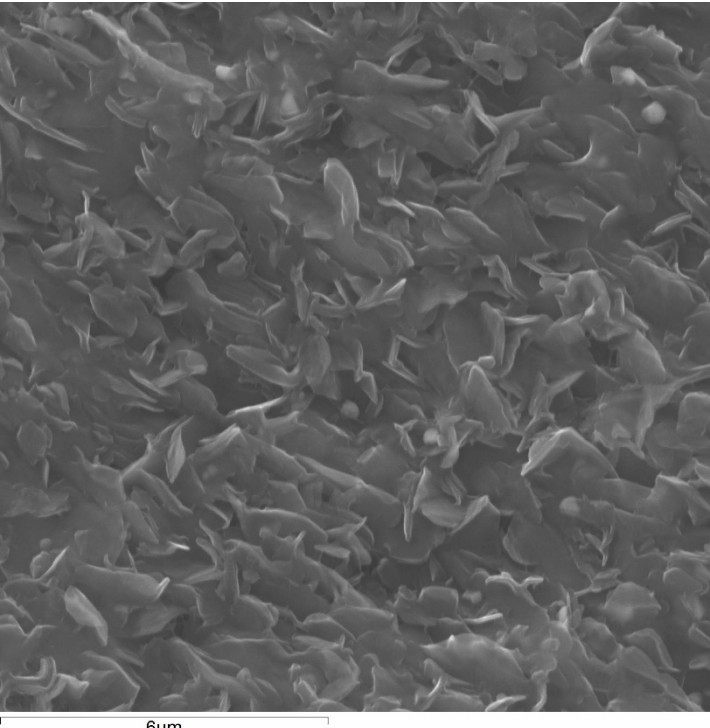

6μm

**Figure 14.** Small flakes of well-crystallized graphite on the surface of a diamond plate exposed to HPHT effects.

Preservation of the diamond core and thickening of the graphite layer with increasing temperature or increase of run duration is evidence of surface graphitization in type 1 diamonds (D1 1427 and G 1582 cm$^{-1}$ bands). Complete transformation of type 1 impact diamonds into graphite may result from accelerated surface graphitization under higher temperatures as the diamond core reduced till disappearance, rather than from a bulk graphitization process. This is possible due to nanostructural features of impact diamond, mainly of type 1, free from graphite: bulk graphitization starts on defects but becomes obstructed by boundaries of nanometer diamond grains.

The dark Popigai diamonds (type 2) exposed to high temperatures and pressures completely transformed into graphite at about 2000 °C. Light-color diamonds with small lonsdaleite percentages (type 1) are more resistant to high temperature than their dark counterparts containing more lonsdaleite and graphite (type 2). Graphitization in graphite-bearing diamonds of type 2 starts at 150–200 °C lower temperatures than in light-color varieties. Samples of type 2 diamond become graphitized from the surface, as well as over the entire volume, probably due to the presence of graphite inclusions in the initial samples, which become graphite crystallization centers under high temperatures. Breakage of C–C bonds was found to be the limiting stage of the process [31].

The HPHT treatment of type 2 and intermediate type 3/2 diamonds led to considerable (two- to four-fold) band reduction. At the same time, the fraction of the cubic phase in the diamond/lonsdaleite mixture significantly increased while the fraction of lonsdaleite decreased (from 45% to 7% in run 35 and from 52% to 3% in run 46). Thus, impact diamonds of type 2 exposed to high pressures and temperatures (5.5 GPa and 1800–2200 °C) changed dramatically in the diamond/lonsdaleite ratio as the fraction of lonsdaleite decreased, while the fraction of graphite increased due to newly formed well-crystallized graphite. At the same time, the fraction of cubic diamond (integral Raman intensity ratio) in the diamond-graphite mixture decreased from 3.8 to 0.2, for example, in sample 3. The diamond band in the spectra of this sample broadened and shifted from 1317.2 cm$^{-1}$ before to 1332–1335.8 cm$^{-1}$ after the HPHT run. All these trends show up in a more prominent way at longer run durations, and the relatively short runs we applied may be insufficient for homogeneous transformation.

According to TEM and XRD data, impact diamonds have a complex polycrystalline and twinned structure consisting of nanocrystals [11]. The increasing share of graphite in the samples exposed to HPHT conditions, along with the dramatic change in the lonsdaleite/diamond ratio, would mean that the diamond-lonsdaleite mixture converts to graphite; however, this hypothesis contradicts the increase of cubic diamond. Therefore, conversion to graphite is not proportional to the original diamond/lomsdaleite ratio, and a large fraction of lonsdaleite disappears. This effect may explain the decrease in the lonsdaleite percentage in treated diamonds of types 3/2 and 2.

Cubic diamond may be more resistive to graphitization at high temperatures than lonsdaleite as an independent hexagonal phase, and graphitization in the two phases has different rates. However, the nanocrystalline diamond-lonsdaleite mixture would have disintegrated into nanometer particles, which is not the case. Note that annealed diamonds always retain a minor amount of lonsdaleite as a structure defect (Figure 15), which may be necessary for the stabilization of the diamond cubic structure under the high-temperature conditions.

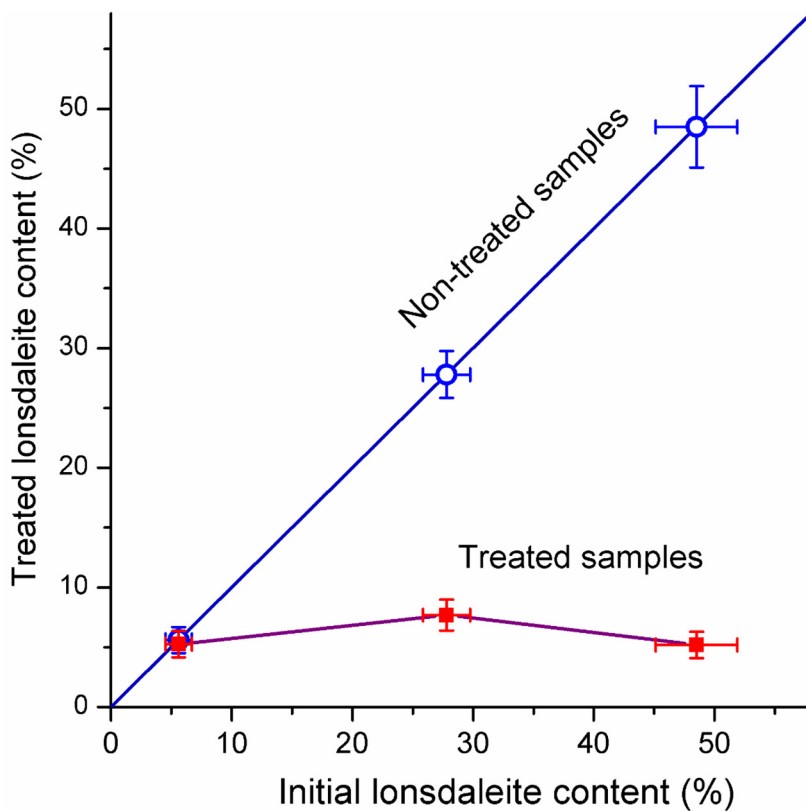

**Figure 15.** Relative percentages of lonsdaleite (*L*) and cubic diamond (100 − *L*) in impact diamonds of types 1 (3.2%–8.0%), 2 (45%–52%), and 3/2 (25.9%–29.7%) before (blue line and rings) and after (purple line and red squares) HPHT runs, according to Raman spectroscopy (Table 1). Points (rings and squares) correspond to average content of lonsdaleite in three samples types.

Otherwise, the lonsdaleite structure in diamond-lonsdaleite impact samples may convert to the structure of cubic diamond under HPHT effects, while diamond undergoes graphitization. Thereby peaks in the Raman spectra become narrower and higher, which is evidence of a more perfect structure in annealed diamond. Graphitization occurring on the surface and propagating inward the type-1 sample, instead of involving the bulk volume, lags behind the lonsdaleite-to-diamond conversion. Thus, lonsdaleite is rather a structure defect than an independent phase of hexagonal diamond.

## 5. Conclusions

Analysis of the results can lead to several important points:

1. The percentages of lonsdaleite from the lonsdaleite/diamond Raman intensity ratio was estimated at 0 to 10% and 40 to 55% in samples of types 1 and 2, respectively, and 20 to 30% in those of intermediate type.

2. For impact diamonds of type 1 (with 0%–10% lonsdaleite content in the lonsdaleite-diamond mixture) under the conditions of the experiments performed, only the process of surface graphitization is observed (bulk graphitization is absent), and they are more resistant to graphitization as compared to impact diamonds of type 2 (with 40%–55% lonsdaleite content).

3. For impact diamonds of type 2 with high content of lonsdaleite and graphite inclusions, the competing process with respect to surface graphitization is bulk graphitization occurring at a much higher rate.

4. The spectra measured sufficient homogeneity of the samples at three or four points, with minor variations of diamond percentage appearing as a small shift within 2 cm$^{-1}$.

5. When type 2 impact diamonds are annealed, the share of cubic phase increases and the share of hexagonal phase sharply decreases from 45%–52% up to 3.3%–7.1% (Table 1, Figure 15).

6. During annealing of type 1 impact diamonds, the share of hexagonal phase relative to the share of cubic phase practically does not change and remains at the same small level from 3.2%–8.0% up to 4.5%–6.0% (Table 1, Figure 15).

One can suppose that lonsdaleite domains are less stable during annealing than in the cubic phase, and in the absence of bulk graphitization in type 1 diamonds, the rate of transformation of the hexagonal phase into the cubic phase is higher than the graphitization rate. Change of the lonsdaleite/diamond ratio during annealing of type 2 diamonds in favor of the cubic phase may confirm the assumption that the lonsdaleite is a defect of the diamond lattice (as stacking faults) rather than an independent phase of hexagonal diamond.

**Author Contributions:** Conceptualization, writing, original draft preparation, A.C. (Anatoly Chepurov), S.G. (Sergey Goryainov); XRD analysis, S.G. (Sergey Gromilov), analysis of diamonds, A.C. (Aleksey Chepurov), V.A.; resources, V.S.; funding acquisition, project administration, N.P.; methodology, E.Z., Z.K. All authors have read and agreed to the published version of the manuscript.

**Funding:** The study was supported by grant No. 075-15-2020-781 from the Ministry of Science and Higher Education of the Russian Federation.

**Data Availability Statement:** Not applicable.

**Conflicts of Interest:** The authors declare no conflict of interest.

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
