# Peer review of "HPHT-Treated Impact Diamonds from the Popigai Crater (Siberian Craton): XRD and Raman Spectroscopy Evidence"

_minerals, doi:10.3390/min13020154_

Round 1

Reviewer 1 Report

The manuscript is about XRD and Raman Spectroscopy Evidence on HPHT-Treated Impact Diamonds from the Popigai Crater (Siberian Craton). The manuscript is well-written, and it gives important information that enriches the connected literature. 

I suggest to the authors pay attention to the results. Many of the differences reported in the Raman spectra are under the instrument resolution. So, the authors should correct their data presentation and pay more attention to the conclusion.  

Author Response

The text has been corrected in accordance with the comments.

Author Response

Thank you, we have updated the text.

Reviewer 3 Report

The authors carried out high pressure and high temperature treatment on diamonds from the Popigai impact crater together with the Raman spectroscopy and X-ray diffractometry. The light-color diamonds are more resistant to HPHT effects than dark diamonds and propose that lonsdaleite is more likely a structure defect than a separate hexagonal phase.  This manuscript is well-written, and the results is properly discussed. As a result, I would suggest publication after some possible changes in the discussion.

In the introduction part, more description of high pressure high temperature (HPHT) treatment is needed. HPHT is a novel technique in materials synthesizing, especially high temperature superconductors(Science advances 4 eaau0192 2018), organic superconductors(Phys. Rev. B 96, 224501, 2017),  black phosphorus, etc.  

In Fig 3, is there any EDX result about the composition differences between the colorless and yellowish diamonds?

Author Response

Thanks for the valuable comments. The HP-HT method is well developed for diamonds and has been shown in many publications. In this article, it is described in sufficient volume. For other materials, the method must be adapted. We are ready to contact the reviewer on these issues. All fixes are highlighted in yellow.
